# Implementation of a Fuel Estimation Algorithm Using Approximated Computing

**Imed Ben Dhaou** [1,2,3]

1   Department of Computer Science, Hekma School of Engineering, Computing, and Informatics,
    Dar Al-Hekma University, Jeddah 22246-4872, Saudi Arabia; imed.bendhaou@utu.fi
2   Department of Computing, University of Turku, FI-20014 Turku, Finland
3   Department of Technology, Higher Institute of Computer Sciences and Mathematics, University of Monastir,
    Monastir 5000, Tunisia

**Abstract:** The rising concerns about global warming have motivated the international community to take remedial actions to lower greenhouse gas emissions. The transportation sector is believed to be one of the largest air polluters. The quantity of greenhouse gas emissions is directly linked to the fuel consumption of vehicles. Eco-driving is an emergent driving style that aims at improving gas mileage. Real-time fuel estimation is a critical feature of eco-driving and eco-routing. There are numerous approaches to fuel estimation. The first approach uses instantaneous values of speed and acceleration. This can be accomplished using either GPS data or direct reading through the OBDII interface. The second approach uses the average value of the speed and acceleration that can be measured using historical data or through web mapping. The former cannot be used for route planning. The latter can be used for eco-routing. This paper elaborates on a highly pipelined VLSI architecture for the fuel estimation algorithm. Several high-level transformation techniques have been exercised to reduce the complexity of the algorithm. Three competing architectures have been implemented on FPGA and compared. The first one uses a binary search algorithm, the second architecture employs a direct address table, and the last one uses approximation techniques. The complexity of the algorithm is further reduced by combining both approximated computing and precalculation. This approach helped reduce the floating-point operations by 30% compared with the state-of-the-art implementation.

**Keywords:** FPGA; eco-driving; floating-point arithmetic

## 1. Introduction

The transportation sector is experiencing a paradigm shift thanks to the fast development in information and communication technologies (ICT). Sustainability and a multitude of other factors have contributed to the establishment of transportation 4.0. It has been argued in many published reports that the legacy transportation system is inefficient, polluting, and unsafe [1–3]. To remedy the problems associated with road traffic, the intelligent transportation system based on dedicated short range communication (DSRC), has been proposed with the following chief aims [4]: reduce congestion, increase road safety, improve drive experience, lower greenhouse gas emission, and make the transportation more efficient. In [5], the authors proposed an eco-routing system based on vehicle-to-infrastructure (V2I). The energy consumption on a given road is transmitted to a road-side unit (RSU) and forwarded to the traffic management center (TMC). Drivers use this information to find fuel-efficient routes. This solution necessitates a strong ICT infrastructure, which can be prohibitive. The same design principle has been advocated in [6].

The revolution in communication, embedded systems, and the associated disruptive technologies have contributed to the realization of the fourth industrial revolution, commonly known as industry 4.0. The smart city is yet another concept that emerged with the development of the Internet of Things, as well as machine learning techniques. As pictured

in Figure 1, smart mobility is one of the pillars in the smart city [7]. In the EU model for the smart city, the smart mobility indicator includes safety, sustainability, and innovation.

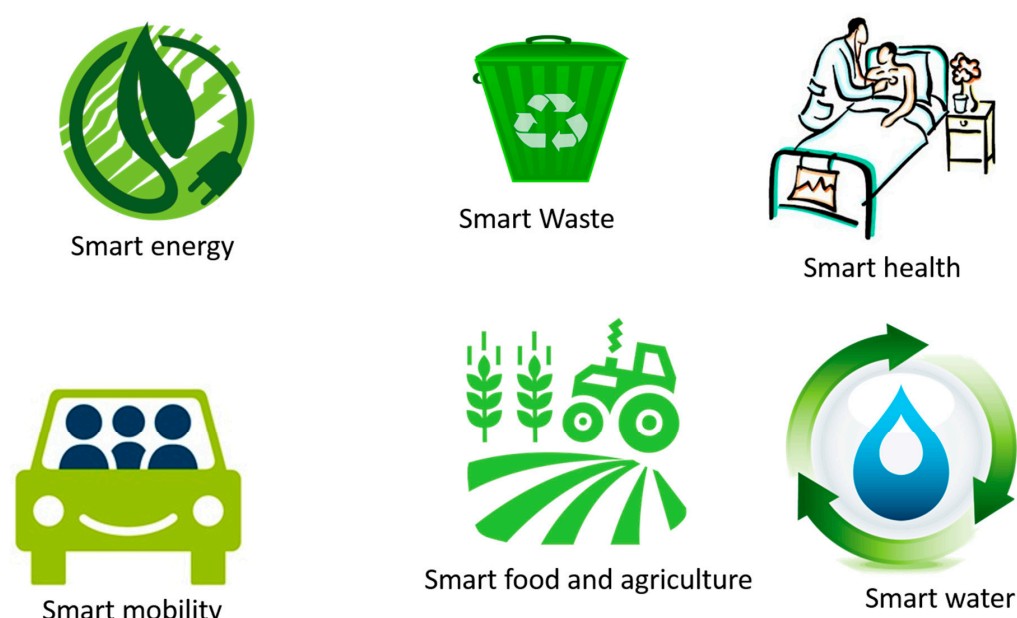

**Figure 1.** Pillars of the smart city framework.

In academia, reducing car emissions has been the focus of intensive work. Many parameters impact car fuel consumption (load, tire pressure, road, weather, vehicle age, etc.). It has been demonstrated through intensive experiments that driving style substantially impacts fuel consumption. Three driving styles have been investigated in [8]. It was found that economic driving style reduces fuel consumption by 21% compared to dynamic driving.

In the literature, several algorithms have been devised for the estimation of fuel consumption [9–12]. Real-time implementation of fuel estimation algorithms has received scant attention. This work is an extension of previously published work [9,13]. The contributions reported in this work are the following:

- Reducing the computational complexity by 66% using high-level transformation techniques;
- Devising two techniques for computing the RPM: binary searching based on a direct-addressable table and approximation algorithm;
- Implementation of the devised architecture using both an IP method and a high-level synthesis tool (GAUT).

The rest of the paper is organized as follows. Section 2 compares our work with existing techniques. Section 3 reviews the fuel estimation algorithm described in [9]. Section 4 describes techniques to reduce the computational complexity of the fuel estimation algorithm and elaborates three hardware architectures. Section 5 reports the implementation results. Section 6 concludes the paper.

## 2. Related Work

In the last decade, numerous fuel estimation algorithms have been proposed. The authors of [14] elaborated an algorithm using a power-based model. The algorithm requires instantaneous values for the acceleration and speed; consequently, it cannot be used for eco-routing.

An Android application was devised in [15]. The app reads vehicle parameters through on-board diagnostics parameter ID (OBDII) interface. The system uses artificial intelligence techniques to provide the driver with eco-driving tips.

Using the Willan's internal combustion engine, the author of [9] devised a non-iterative fuel estimation model. The technique devised in [16] for the vehicle routing problem (VRP) is determined by the comprehensive modal emission model (CMEM). CMEM requires both speed and acceleration to estimate the fuel consumption. The engine RPM was fixed to 2800 for a passenger vehicle and 2400 for a truck. The authors of [13] envisioned an RPM algorithm and designed a hardware architecture using floating point arithmetic for the implementation of the fuel estimation algorithm.

Approximated computing is a new design technique that has been conceived to reduce the power consumption or increase the speed of VLSI circuits. Floating-point arithmetic consumes more area, is slower, and is more power-hungry compared with fixed-point arithmetic. Fortunately, approximated computing have also been shown to substantially reduce delay and power consumption [17].

## 3. Real-Time Fuel Estimation Algorithm

### 3.1. Preliminaries

Table 1 lists the parameters along with their typical values used to elaborated on the architecture.

**Table 1.** Parameters list with their their typical values.

| Parameter | Explanation | Unit | Typical Value/Range |
|:---:|:---:|:---:|:---:|
| $d$ | Distance | meter | NA |
| $t$ | trip time | second | NA |
| RPM | Rotations per minute | dimensionless | 0–7000 |
| $m_v$ | Vehicle's mass | kg | 750 |
| $m_l$ | Vehicle's load | kg | 600 |
| $B$ | Diameter of the cylinder (in piston) | m | 0.067 |
| $S$ | Engine stroke | m | 0.067 |
| $z$ | number of cylinders | dimensionless | 4–8 |
| $A_f$ | Frontal area of a vehicle | m$^2$ | 1.97–3.2 |
| $c_d$ | drag coefficient | dimensionless | 0.29–0.32 |
| $c_r$ | A rolling friction coefficient | dimensionless | NA |
| $t_{trac}$ | percentage of traction time | dimensionless | 60% |
| $\rho_a$ | Ambient air density | kg/m$^3$ | 1–1.3 |
| $e_{gb}$ | efficiency index of the gearbox | dimensionless | 0.92–0.95 |
| $P_{0,gb}$ | the gearbox power loss | W | 3% of the rated power of the gear box |
| $N_{stops}$ | Frequency of stopping | dimensionless | NA |
| $H_l$ | the fuel's lower heating value | J/kg | $43.510^6$ (for RON—95 gasoline) |
| $\rho_f$ | Density of the fuel | kg/L | 0.75 (for RON—95 gasoline) |
| $g$ | earth acceleration | m/s$^2$ | 9.83 |
| $F_{trac}$ | attractive force | N | NA |
| $\zeta_e$ | engine efficiency | dimensionless | NA |

Subsequently, we review the equations used to elaborate the macroscopic fuel estimation algorithm described in [9,18,19].

Figure 2 summarizes the forces exerted on a vehicle in an acceleration mode. Those forces are: the traction force ($\vec{F}_t(t)$), the aerodynamic friction($\vec{F}_a(t)$), the rolling friction ($\vec{F}_r(t)$), the gravity force($\vec{F}_g(t)$), and the disturbance force ($\vec{F}_d(t)$). The relation between those forces is described in (1) [19].

$$m_v \vec{a} = \vec{F_t}(t) - \left( \vec{F_a}(t) + \vec{F_r}(t) + \vec{F_g}(t) + \vec{F_d}(t) \right), \tag{1}$$

where is $m_v$ denotes the mass of the vehicle's body, and $\vec{a}$ represents the acceleration of the vehicle.

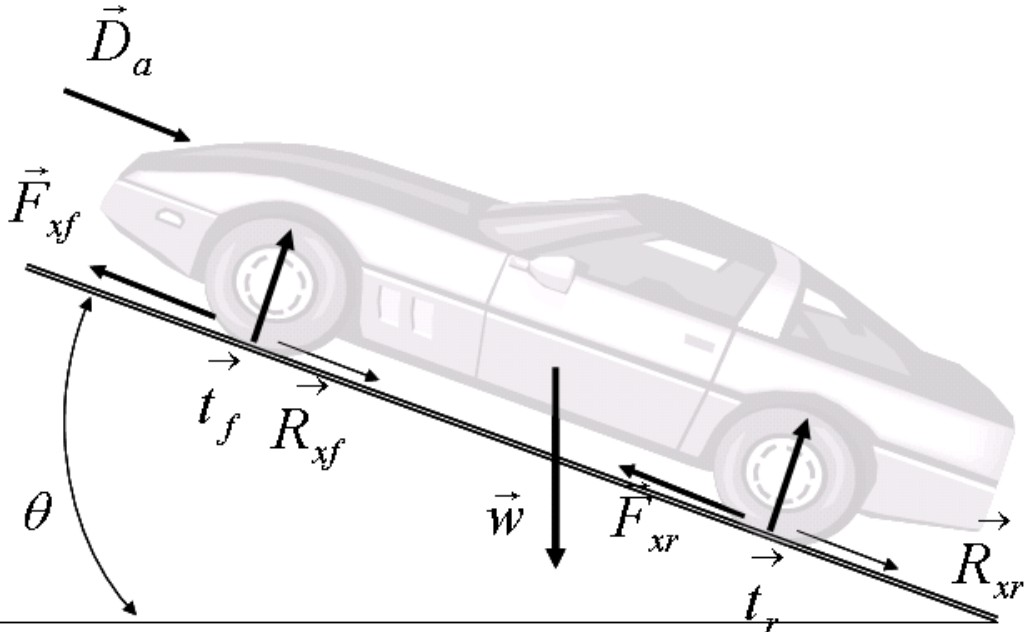

**Figure 2.** Major forces acting on an accelerated vehicle along a ramp.

In traction mode, the distance traveled by the vehicle is determined using (2).

$$x_{total} = \int_{t \in trac} v(t) dt. \tag{2}$$

Equations (3) and (4) are, respectively, used to estimate the average value of the speed and the acceleration.

$$\bar{v}_i = \frac{v_i + v_{i-1}}{2}. \tag{3}$$

$$\bar{a}_i = \frac{v_i - v_{i-1}}{h}. \tag{4}$$

Given the air density ($\rho_a$), the vehicle frontal area ($A_f$), the aerodynamic drag coefficient ($c_d$), and the rolling friction coefficient ($c_r$), (5)–(8) are used to calculate the average value of the traction force.

$$\bar{F}_t = \bar{F}_a + \bar{F}_r + \bar{F}_m, \tag{5}$$

where $\bar{F}_m$ is the acceleration force.

$$\bar{F}_a = \frac{1}{x_{tot}} \frac{1}{2} \rho_a A_f c_d \sum_{i \in trac} \bar{v}_i^3 h. \tag{6}$$

$$\bar{F}_r = \frac{1}{x_{tot}} m_v g c_r \sum_{i \in trac} \bar{v}_i h. \tag{7}$$

$$\bar{F}_m = m_v \frac{1}{x_{tot}} \sum_{i \in trac} \bar{a}_i \bar{v}_i h. \tag{8}$$

The mechanical power is determined from the traction as shown in (9).

$$\bar{P}_{trac} = \frac{\bar{F}_t \bar{v}}{t_{trac}}.$$

$$(9)$$

The gearbox input power is calculated from (10).

$$P_{i,gb} = \frac{1}{e_{gb}} \left( \bar{P}_{trac} + P_{0,gb} \right),$$

$$(10)$$

where $e_{gb}$ is the efficiency of the gearbox, and $P_{0,gb}$ the idle power of the gearbox at a given engine speed.

The additional energy consumed after each stop is approximated using (11).

$$E_{start} \approx \frac{1}{2} v_0^2 m_v,$$

$$(11)$$

where $v_0$ is the attained speed from the standing mode.

The mean value of the fuel power consumption given by (12).

$$\bar{P}_{fuel} = t_{trac}(P_{start} + \bar{P}_{trac}) \left[ \frac{p_{me} + p_{me0}(c_m)}{\zeta_e p_{me}} \right],$$

$$(12)$$

where $\zeta_e$ = the engine efficiency parameter, $p_{me}$ = fuel mean pressure, $p_{meo}$ = pressure losses inside the engine, and $P_{start}$ = power consumed when the vehicle is accelerating from a standstill to a given speed. Both $p_{meo}$ and $\zeta_e$ can be obtained from the engine map.

The fuel flow is estimated using (13).

$$V_f = \bar{P}_{fuel} / \left( H_l \cdot \rho_f \right),$$

$$(13)$$

where $H_l$ (J/kg) is the fuel's lower heating value, and $\rho_f$ (kg/l) is the fuel density.

### 3.2. Algorithm for Fast Fuel Estimation

To simplify the calculations, the author of [9] used the driving cycle to estimate the fuel consumption. The known driving modes are MVEG-95 (Motor Vehicle Emissions Group), ECE (European Cycle Emission), and EUDC (Extra-Urban Driving Cycle) [19]. The steps are described in Algorithm 1.

---

**Algorithm 1** Fuel estimation algorithm elaborated in [9].

1: **procedure** FUEL ESTIMATION($d$, $t$, RPM, $m_v$,$B$, $S$, $z$, $A_f$, $c_d$, $c_r$, $t_{trac}$, $\rho_a$, $e_{gb}$, $P_{0,gb}$, $N_{stops}$, $H_l$, $\rho_f$, $g$, $m_l$)

2: $\quad \bar{v} \leftarrow \frac{d}{t}$

3: $\quad \omega_e \leftarrow RPM(\bar{v})$

4: $\quad M \leftarrow m_v + m_l$

5: $\quad h_1 \leftarrow \frac{1}{x_{tot}} \sum_{i \in trac} \bar{v}_i^3 h$

6: $\quad h_2 \leftarrow \frac{1}{x_{tot}} \sum_{i \in trac} \bar{v}_i h$

7: $\quad h_3 \leftarrow \frac{1}{x_{tot}} \sum_{i \in trac} \bar{a}_i \bar{v}_i h.$

8: $\quad F_{trac} \leftarrow h_1 \frac{1}{2} \rho_a A_f c_d + h_2 \cdot M \cdot g \cdot c_r + h_3 M$

9: $\quad P_{trac} \leftarrow \frac{F_{trac} \cdot \bar{v}}{t_{trac}}$

10: $\quad P_{i,gb} \leftarrow \frac{P_{trac} + P_{0,gb}}{e_{gb}}$

11: $\quad P_{start} \leftarrow \frac{4.5 \cdot M}{N_{stops} \cdot \bar{v}}$

12: $\quad c_m \leftarrow \frac{2 \cdot S \cdot \omega_e}{60}$

13: $\quad p_{me} \leftarrow \frac{16 \cdot P_{i,gb}}{z \pi B^2 c_m}$

14: $\quad \zeta_e \leftarrow \frac{0.4 \cdot p_{me} \cdot 10^{-5}}{p_{me} \cdot 10^{-5} + 1.6}$

15: $\quad P_{fuel} = \frac{t_{trac} \cdot (P_{i,gb} + P_{start})}{\zeta_e}$

16: $\quad V_f \leftarrow \frac{P_{fuel}}{H_l \cdot \rho_f}$

17: $\quad \hat{f}_c \leftarrow \frac{V_f \cdot 10^5}{\bar{v}}$

18: **end procedure**

---

## 4. Optimized Hardware Architecture

The hardware architecture for implementing the fuel estimation algorithm needs to have the following features: (1) an RPM unit that determines the engine rotation per minute given the driving speed; (2) a functional unit for computing $h_1$, $h_2$, and $h_3$; (3) a hardware module for calculating $F_{trac}$, $P_{trac}$, $P_{i,gb}$, $P_{start}$,$P_{fuel}$, $V_f$, $\hat{f}_c$; and (3) a memory unit for storing the constants used by the precedent unit.

To reduce the computational complexity of the fuel estimation algorithm, the following techniques can be used efficiently. A predefined driving mode can be utilized to estimate $h_1$, $h_2$, and $h_3$. Table 2, lists the values of $h_{1,2,3}$ for the three driving cycles.

**Table 2.** Statistical values of $h_1$, $h_2$, and $h_3$.

| Driving Cycle | $h_1$ | $h_2$ | $h_3$ |
|:---:|:---:|:---:|:---:|
| MVEG-95 | 319 | 0.856 | 0.101 |
| ECE | 82.9 | 0.81 | 0.126 |
| EUDC | 455 | 0.88 | 0.086 |

The computational complexity can be further reduced by precomputing the first term of $P_{trac}$, that is, the quantity $h_1 \frac{1}{2} \rho_a A_f c_d$. Table 3 reports the precomputed values for the three driving cycles for air density $\rho_a = 1.293 \text{ kg/m}^3$, $c_d = 0.312$, $A_f = 2.06 \text{ m}^3$.

**Table 3.** Precomputed value of $K_1 = h_1 \frac{1}{2} \rho_a A_f c_d$ in N.

| MVEG-95 | ECE | EUDC |
|:---:|:---:|:---:|
| 132.38 | 34.44 | 189.05 |

It is further possible to reduce the computational cost of tractive force, $F_{trac}$, by the transformation shown in (14):

$$F_{trac} = K_1 + M(h_2 \cdot g \cdot c_r + h_3) = K_1 + K_2 \cdot M, \tag{14}$$

where $K_1$ and $K_2$ are constants that depend on the driving cycle. Tables 2 and 4 show the value of, respectively, $K_1$ and $K_2$ for the three driving cycle modes.

**Table 4.** Precomputed value of $K_2 = h_2 \cdot g \cdot c_r + h_3$ for $c_r = 8.63 \times 10^{-3}$.

| MVEG-95 | ECE | EUDC |
|---|---|---|
| 0.173 | 0.194 | 0.160 |

Having performed the rearrangements, the number of multiplications is reduced from 3 to 1. The required number of arithmetic operations before and after optimization is summarized in Table 5. The control data-flow graph of the original and refined algorithm is shown in Figure 3. The control data-flow graph can be further improved by inserting pipeline latches, which reduce the critical path delay to one arithmetic unit, that is $\tau = \max(t_{add}, t_{divider}, t_{multiplier})$, where $t_{add}, t_{divider}, t_{multiplier}$ are, respectively, the critical path delay of a floating point adder, divider, and multiplier.

**Table 5.** Arithmetic cost of the fuel estimation algorithm.

| Operation | Before Transformation | After Transformation | Savings in (%) |
|---|---|---|---|
| Addition | 6 | 4 | 33.3 |
| Multiplication | 11 | 8 | 27.7 |
| Division | 4 | 4 | 0 |

The fuel estimation algorithm requires as input the RPM value ($\omega_e$). The closed-form expression to compute $\omega_e$ is shown in (15).

$$\omega_e = \frac{\bar{v}\gamma_A\gamma_G}{\pi D}, \tag{15}$$

where $D$ is the diameter of the wheel expressed in meters, $\gamma_A$ is the axle ratio, and $\gamma_G$ is the gearbox ratio.

In [19], the author proposed an approximation formula to compute the engine RPM, which is described in (16):

$$\omega_e = \frac{2 \cdot \bar{v}\gamma_i}{D}, \tag{16}$$

where $\gamma_i$ is the gear ratio. The RPM unit can be implement in one of the following ways: a look-up table that stores a pre-calculated value or a datapath unit that computes $\omega_e$ using (16). To verify the accuracy of (16), the RPM and speed have been measured using a sedan vehicle. Table 6 compares the measured RPM to the approximated one using (16). The results shows that the approximation error has an acceptable accuracy.

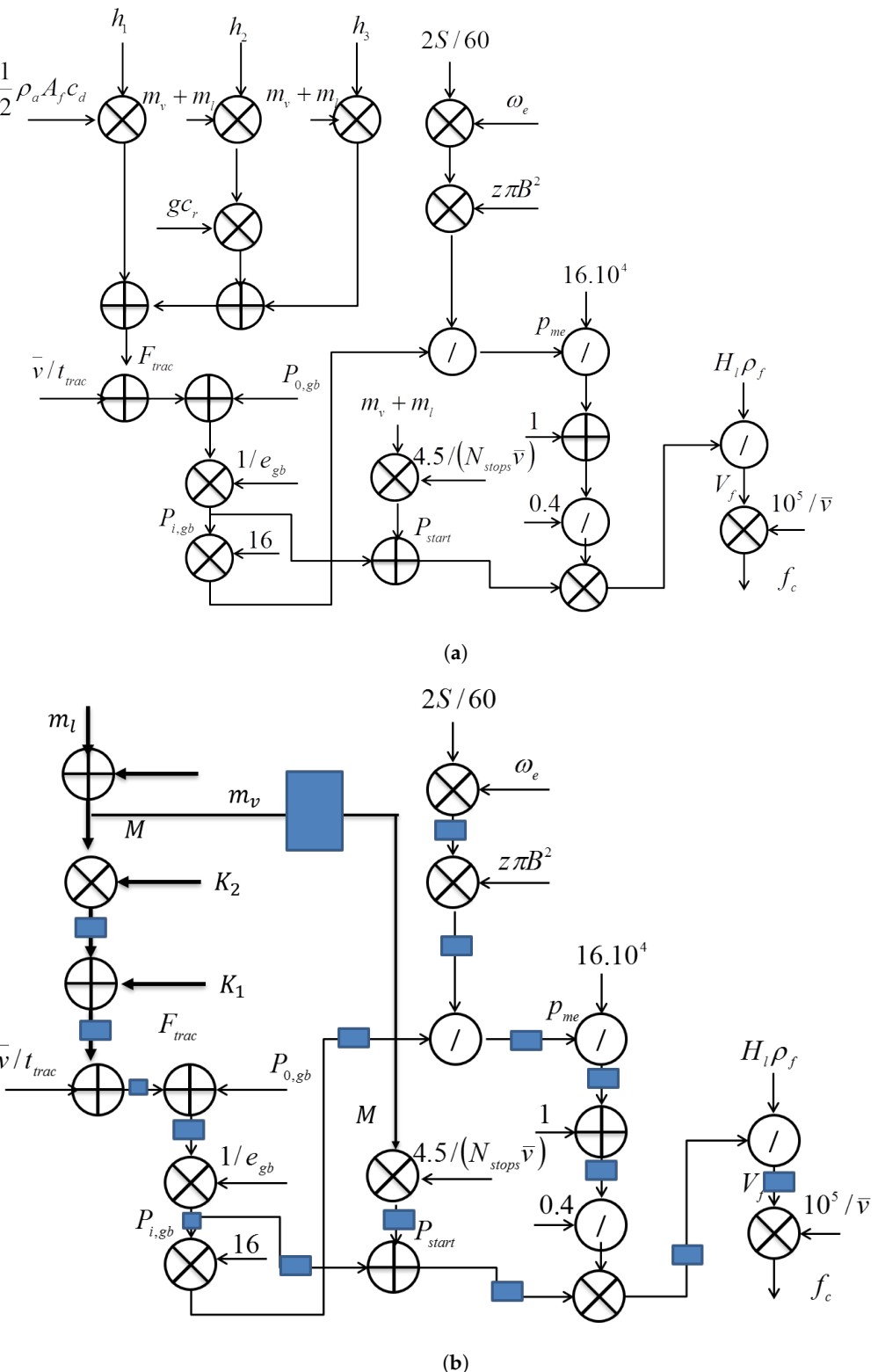

**Figure 3.** Two methods for the calculation of the RPM: (**a**) CDFG for the unoptimized fuel estimation algorithm; (**b**) pipelined and optimized CDFG for the refined fuel estimation algorithm.

**Table 6.** Comparison between measured and estimated RPM.

| Speed (kmph) | Gear Position | Gear Ratio | Measured $\omega_e$ (tr/min) | Estimated $\omega_e$ | Error in % |
|---|---|---|---|---|---|
| 20 | 2nd | 1.93 | 1500 | 1140 | 31.5 |
| 30 | 2nd | 1.93 | 2000 | 1710 | 17 |
| 40 | 3rd | 1.29 | 1800 | 2116 | 15 |
| 50 | 3rd | 1.29 | 2000 | 2645 | 24.38 |
| 60 | 3rd | 1.29 | 2500 | 3174 | 21.23 |
| 70 | 4th | 1 | 2200 | 2870 | 23.3 |
| 80 | 4th | 1 | 2500 | 3280 | 23.7 |
| 90 | 5th | 0.68 | 2200 | 2509 | 12.3 |
| 100 | 5th | 0.68 | 2500 | 2788 | 10.3 |
| 110 | 5th | 0.68 | 2700 | 3067 | 12 |
| 120 | 5th | 0.68 | 3000 | 3346 | 10.3 |
| 130 | 5th | 0.68 | 3000 | 3625 | 17.24 |

The RPM unit designed using a look-up table requires the implementation of a searching algorithm. The algorithm takes as input the average speed and returns the engine RPM ($\omega_e$). The known searching algorithms are: linear, binary, hash table, and direct-table. The complexity of those algorithms is summarized in Table 7. Hardware implementation of the search algorithm. For the fuel estimation algorithm, the direct address table is the most appropriate way for searching the engine RPM, as the size of the table is small. The Algorithm 2 computes the engine RPM using the direct address table.

**Table 7.** Complexity-based search techniques.

| Algorithm | Linear Search | Binary Search | Hash Table | Direct Address Table |
|---|---|---|---|---|
| Complexity | $\mathcal{O}(n)$ | $\mathcal{O}(\log n)$ | $\mathcal{O}(1)$ | $\mathcal{O}(1)$ |

The data-flow graph of the RPM unit using the approximating equation, the search algorithm using the direct address table, and the binary search algorithm are pictured in Figure 4.

---

**Algorithm 2** Pseudo-code for the RPM calculation unit.

---

1: **procedure** RPM-2($\bar{v}$, **TRPM**, **Tspeed**)
2:     I $\leftarrow (0.1 \cdot \bar{v}) - 1$
3:     $\omega_e \leftarrow$ **TRPM**(I)
4:     **return** $\omega_e$
5: **end procedure**

---

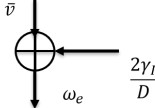

**(a)**

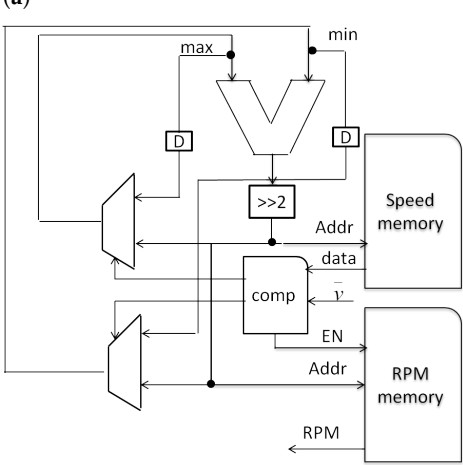

**(b)**

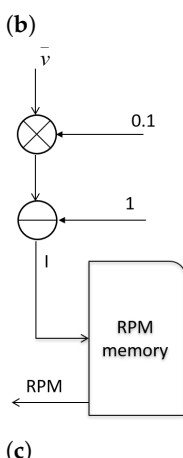

**(c)**

**Figure 4.** Three methods for the calculation of the RPM: (**a**) DFG for the RPM calculation unit using approximated equation; (**b**) RPM unit calculation using a binary search algorithm; (**c**) RPM unit calculation using a direct address table.

## 5. Implementation Results

Three types of architecture have been implemented using FPGA technology. Those architectures are summarized in Table 8. The architecture ArchApproxRM uses an approximation formula to compute $\omega_e$. The binary search algorithm is used by the architecture ArchBinaryRPM. The last one uses a direct address table.

**Table 8.** Features of the hardware architectures.

| Architecture | ArchApproxRPM | ArchBinaryRPM | ArchDirectRPM |
|:---:|:---:|:---:|:---:|
| Feature | Approximation | Binary Search | Direct |

For automated architecture synthesis of the RPM unit, the academic high-level synthesis tool (GAUT) has been used [20]. The design flow using GAUT is shown in Figure 5. Comparison between three architectures for the implementation of the RPM calculator is presented in Table 9. From the presented results, it is clear the the approximation method consumes fewer resources as compared with the two others.

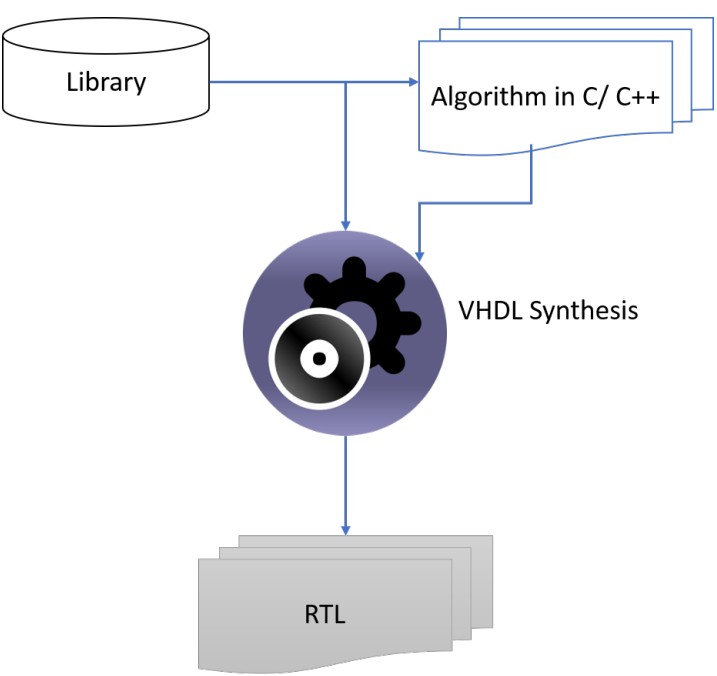

**Figure 5.** High-level synthesis using GAUT.

**Table 9.** Synthesis results for the RPM calculator.

| Architecture | ArchBinaryRPM | | ArchDirectRPM | | Approximation | |
|---|---|---|---|---|---|---|
| **Block Name** | **Size** | **Number** | **Size** | **Number** | **Size** | **Number** |
| Single-port ROM | $32 \times 12$ | 1 | $32 \times 12$ | 1 | 0 | 0 |
| Adders | 5-bits | 11 | 0 | 0 | 0 | 0 |
| Comparators | 9-bits | 24 | 0 | 0 | 0 | 0 |
| Multiplexers | 2-to-1 | 44 | 0 | 0 | 0 | 0 |
| Multiplier | 0 | 0 | $10 \times 10$ | 1 | $10 \times 10$ | 1 |

The data path of the fuel estimation algorithm necessitates the implementation of floating-point arithmetic. This is due to the high-dynamic range of the coefficients used in the fuel estimation algorithm. Floating-point arithmetic is supported in VHDL 2008 operations [21]. Furthermore, a number of vendors offer IP for floating point operations. To select the suitable method for the realization of the data path, both the floating-point package and the IP provided by Xilinx ISE tool were implemented and tested using Virtex-6 FPGA. The core is reconfigurable and can be used to design adder, multiplier, absolute value, exponential function, square-root, conversion between fixed and floating-point, natural logarithm, and accumulator [22].

The comparison presented in Table 10 favors the IP over the floating-point package. The critical delay for the data path is 4 ns, which is nearly 17.8 times less than the one proposed in [13]. The proposed architecture can be used for eco-routing. The circuit can be further optimized by fine-tuning the floating-point arithmetic. Furthermore, the fuel estimation model does not consider factors such as driving comfort and weather conditions. A more detailed investigation will be conducted from both an algorithm standpoint as well as a hardware implementation standpoint.

**Table 10.** Comparison between a commercial IP and the floating point package.

| Operator | $F_{max}$ (IP) | $F_{max}$ (fp Package) |
|---|---|---|
| Divide | 237 MHz | 13.6 MHz |
| Multiplier | 245 MHz | 38.2 MHz |
| Adder | 236 MHz | 22.8 MHz |

## 6. Conclusions

Approximation computing is a very attractive design techniques for reducing the implementation complexity of DSP algorithms. In this paper, high-level transformation techniques were used to implement a fuel estimation algorithm for a passenger car. Three architectures were devised and implemented on FPGA technology. High-level synthesis using GAUT was employed to design the RPM calculation unit. Pre-calculation combined with approximated computing was exercised to further reduce the floating-point operations. The results show that the architecture that uses approximated multiplication for the computation of the engine RPM consumes fewer hardware resources compared with the binary-based searching method. Furthermore, the control data-flow graph was pipelined to reduce the critical path delay. The synthesis results were determined using a commercial IP. The maximum operating frequency of the data path is 236 MHz, which is dictated by the floating-point adder.

**Funding:** This research/project was funded by the Vice Presidency for Graduate Studies, Business, and Scientific Research (GBR) at Dar Al Hekma University, Jeddah, under grant no. (RFC/21-22/006). The author, therefore, acknowledges with thanks GBR for technical and financial support.

**Institutional Review Board Statement:** The study was conducted according to the guidelines of the Declaration of Helsinki, and approved by the Institutional Review Board (or Ethics Committee) of Dar Al-Hekma University.

**Conflicts of Interest:** The authors declare no conflict of interest.

## Abbreviations

The following abbreviations are used in this manuscript:

| | |
|---|---|
| MVEG | Moving Vehicle Emission Group |
| NEDC | The New European Driving Cycle |
| ECE | European Cycle Emission |
| EUDC | Extra-Urban Driving Cycle |
| CMEM | Comprehensive Modal Emission Model |
| IP | Intellectual Property |
| DFG | Data-Flow Graph |

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
