# Peer review of "Implementation of a Fuel Estimation Algorithm Using Approximated Computing"

_jlpea, doi:10.3390/jlpea12010017_

Round 1

Reviewer 1 Report

This paper elaborates a VLSI architecture for the fuel estimation algorithm. The comments are listed:

1) There are half of content for the background in the current abstract. The abstract should be modified to show your highlights.

2) More reviews and discussions can be added in introduction, such as the latest researches, which can be therefore led to your contributions.

3) What about the effects by your fuel consumption estimation method? It can be enriched.

4) There are some typos for “??”, such as on line 44 of page 2. Please check.

5) More validations can be added for comprehensive evaluation, to show your advantages for your method.

Author Response

Response to Reviewer 1 Comments

Point 1: There are half of content for the background in the current abstract. The abstract should be modified to show your highlights..

Response 1: The author is grateful to the comment provided by the reviewer. I have revised the abstract and addded substantial text (colored in blue) to show the approach as well as the results.

Point 2: More reviews and discussions can be added in introduction, such as the latest researches, which can be therefore led to your contributions.

Response 2: Many thanks for brining this point out. I have added 7 more references to enrich the related works (based on the reviewer 1 and reviewer 2 comments). The related works now included recent techniques applied for fuel estimation. Also, approches for approximated computing applied to the floating-point arithmetic.

Point 3: What about the effects by your fuel consumption estimation method? It can be enriched.

Response 3:  Fuel estimation model that is proposed in earlier work and extended here allows for eco-routing. In addition, the algorithm should be VLSI friendly. Those points have been highlighted in the main text.

Point 4: There are some typos for “??”, such as on line 44 of page 2. Please check.

Response 4: Sorry for the typos.  I have fixed those missing references.

Point 5 More validations can be added for comprehensive evaluation, to show your advantages for your method..

Response 4: Thanks for the comments.  I have added the pipelining approach and expended the experimental part with two additional paragraphs.

Reviewer 2 Report

Dear Author,

thank you for getting to know your manuscript, it's a very interesting work. The manuscript deals with issues important for the operation of motor vehicles and their impact on the natural environment - GHG emissions. Eco-driving is one of the directions leading to the reduction of fuel consumption and emissions by motor vehicles. The author presents an interesting approach to engine control and the resulting benefits in the form of lower fuel consumption. Despite the great care of the manuscript, the Author did not avoid editing errors. In addition, it is required to complete some content in the introductory part and the article summary.
Detailed comments are presented below:

  1. The abstract is a bit short, it should be up to 200 words, but not even 100.
  2. please add this thought from lines 14-18 in the next paragraph, where you should also add references to the completed literature. If I can suggest it, I suggest the following items in the field of transport safety [https://doi.org/10.3390/app11114854  https://doi.org/10.3390/en14051476], the phenomenon of transport congestion [https://doi.org/10.20858/sjsutst.2021.112.2.], greenhouse gas emissions [DOI: 10.26552/com.C.2021.4.B265-B277] fuel consumption and eco-driving [https://doi.org/10.1016/j.trpro.2021.07.009 https://doi.org/10.1051/matecconf/201925206009 https://doi.org/10.1016/j.trpro.2019.07.178 https://doi.org/10.1016/j.trd.2009.05.009 https://doi.org/10.1016/j.scs.2014.08.002].
    In line 24 you can add entries for smart-city [10.1109 / SCSP52043.2021.9447376  https://doi.org/10.3390/su13116031].
  3. Shouldn't Figure 1 have a footnote?
  4. It seems to me that there should be a punctuation mark in front of the presented formulas and equations.
  5. The tables in the text are incorrectly edited according to the template.
  6. Check the correctness of the abbreviations ECE and EUDC line 74.
  7. Check if h3 is correct, 3 is the index? Line: 79, eq. (14).
  8. writing the% value is invalid. Line 91.
  9. Place figure 4 correctly in the text.
  10. The Conclusions section is too general, highlight the main strengths of your work here.

Thank you

Author Response

Dear Reviewer

Many thanks for your comments, time, and effort.  I have addressed those comments. All the modifications are highlighted in blue in the revised manuscript.
